# Epidemiology, Clinical Presentation and Treatment of Non-Hepatic Hyperammonemia in ICU COVID-19 Patients

**DOI:** 10.3390/jcm11092592

**Published:** 2022-05-05

**Authors:** Nardi Tetaj, Giulia Valeria Stazi, Maria Cristina Marini, Gabriele Garotto, Donatella Busso, Silvana Scarcia, Ilaria Caravella, Manuela Macchione, Giada De Angelis, Rachele Di Lorenzo, Alessandro Carucci, Alessandro Capone, Andrea Antinori, Fabrizio Palmieri, Gianpiero D’Offizi, Fabrizio Taglietti, Stefania Ianniello, Paolo Campioni, Francesco Vaia, Emanuele Nicastri, Enrico Girardi, Luisa Marchioni

**Affiliations:** 1UOC Resuscitation, Intensive and Sub-Intensive Care, National Institute for Infectious Diseases IRCCS Lazzaro Spallanzani, Via Portuense, 292, 00149 Rome, Italy; giuliavaleria.stazi@inmi.it (G.V.S.); mariacristina.marini@inmi.it (M.C.M.); gabriele.garotto@inmi.it (G.G.); donatella.busso@inmi.it (D.B.); silvana.scarciadaprano@inmi.it (S.S.); ilaria.caravella@inmi.it (I.C.); manuela.macchione@inmi.it (M.M.); giada.deangelis@inmi.it (G.D.A.); rachele.dilorenzo@inmi.it (R.D.L.); alessandro.carucci@inmi.it (A.C.); luisa.marchioni@inmi.it (L.M.); 2Clinical and Research Department of Infectious Diseases, National Institute for Infectious Diseases IRCCS Lazzaro Spallanzani, 00149 Rome, Italy; alessandro.capone@inmi.it (A.C.); andrea.antinori@inmi.it (A.A.); fabrizio.palmieri@inmi.it (F.P.); gianpiero.doffizi@inmi.it (G.D.); fabrizio.taglietti@inmi.it (F.T.); stefania.ianniello@inmi.it (S.I.); paolo.campioni@inmi.it (P.C.); emanuele.nicastri@inmi.it (E.N.); 3Health Direction, National Institute for Infectious Diseases IRCCS Lazzaro Spallanzani, 00149 Rome, Italy; francesco.vaia@inmi.it; 4Scientific Direction, National Institute for Infectious Diseases IRCCS Lazzaro Spallanzani, 00149 Rome, Italy; enrico.girardi@inmi.it

**Keywords:** COVID-19, Coronavirus disease, intensive care unit, non-hepatic hyperammonemia

## Abstract

(1) Background: Although COVID-19 is largely a respiratory disease, it is actually a systemic disease that has a wide range of effects that are not yet fully known. The aim of this study was to determine the incidence, predictors and outcome of non-hepatic hyperammonemia (NHH) in COVID-19 in intensive care unit (ICU); (2) Methods: This is a 3-month prospective observational study in a third-level COVID-19 hospital. The authors collected demographic, clinical, severity score and outcome data. Logistic regression analyses were performed to identify predictors of NHH; (3) Results: 156 COVID-19 patients were admitted to the ICU. The incidence of NHH was 12.2% (19 patients). The univariate analysis showed that invasive mechanical ventilation had a 6.6-fold higher risk (OR 6.66, 95% CI 0.86–51.6, *p* = 0.039) for NHH, while in the multiple regression analysis, there was a 7-fold higher risk for NHH—but it was not statistically significant (OR 7.1, 95% CI 0.90–56.4, *p* = 0.062). Demographics, clinical characteristics and mortality in the ICU at 28 days did not show a significant association with NHH. (4) Conclusions: The incidence of NHH in ICU COVID-19 patients was not low. NHH did not appear to significantly increase mortality, and all patients with non-hepatic hyperammonemia were successfully treated without further complications. However, the pathogenesis of NHH in ICU patients with COVID-19 remains a topic to be explored with further research.

## 1. Introduction

Coronavirus disease 2019 (COVID-19) is an infectious disease caused by a coronavirus discovered in December 2019, named severe acute respiratory syndrome coronavirus-2 (SARS-CoV-2), and is responsible for the current global pandemic that started in March 2020 [1]. SARS-CoV-2 is transmitted from person to person via droplets, contact and also from aerosolized particles [2].

Ammonia (NH_3_) is a final product of protein metabolism that is formed by the action of bacteria on the proteins contained in the intestine and by the hydrolysis of glutamine in the kidneys. Hyperammonemia commonly occurs in patients with hepatic insufficiency—the latter being the primary organ responsible for its degradation into urea through the urea cycle. When ammonium production exceeds the liver’s ability to metabolize it, the elimination of ammonium depends on the kidneys reducing ammonium production and increasing its urinary excretion, as well as on the muscular system and brain, which metabolize ammonium into glutamine [3,4]. Ammonium is known as a potent neurotoxin, although its pathogenesis remains unclear [5]. Therapy for hyperammonemia is often empirical because seeking the cause of the NHH often takes time or even goes undetected.

COVID-19 is a systemic disease, and due to the well-known cytokine storm produced during infection, it is reasonable consider and analyze the hyperammonemia that occurs that is not related to hepatic failure in COVID-19 patients. Non-hepatic hyperammonemia in intensive care patients in the pre-COVID-19 era has already been reported in some studies [6,7].

The medical literature has so far focused on hyperammonemia caused by liver failure in COVID-19 patients [8,9,10], while there are few studies regarding non-hepatic hyperammonemia (NHH)—most of which have a small sample size or are case reports [11,12,13].

The aim of this analysis was to determine the incidence of NHH in ICU COVID-19 patients, as well as NHH-associated risk factors and mortality. We also compared low-NHH and high-NHH, assessed by their risk factors.

## 2. Materials and Methods

### 2.1. Study Design and Participants

This study was conducted at the National Institute for Infectious Disease (INMI) “Lazzaro Spallanzani” in Rome, Italy which is a third-level COVID-19 center with a 200-bed hospital for infectious disease with a 50-bed ICU.

This prospective observational cohort study included adult patients with respiratory failure due to COVID-19 hospitalized in our Intensive Care Unit (ICU) from 1 October 2020 to 31 December 2020. We excluded from the study patients with chronic or acute liver disease. We included in the NHH group COVID-19 patients with hyperammonemia during the first 2 weeks of ICU admission. All patients were tested twice a week for the first 2 weeks after entering the ICU; otherwise, if they had hyperammonemia, they were tested every day until resolution. Non-hepatic hyperammonemia (NHH) is defined as blood ammonia levels >50 µmol/L in COVID-19 patients without a history of liver disease. We defined ammonia blood levels as low-NHH = 55–100 µmol/L and high-NHH > 100 µmol/L, as reported in the literature [14]. Blood samples were stored at 4°C or wrapped in ice packs and analyzed within 1 h after collection. COVID-19 was diagnosed based on the WHO interim guidance [1].

### 2.2. Data Collection

The data collected included age, gender, body mass index (BMI), sequential organ failure assessment score (SOFA) and acute physiology and chronic health evaluation II score (APACHE II) at ICU admission; pre-ICU hospitalization; PaO_2_/FiO_2_ at ICU admission; invasive mechanical ventilation; and ICU length of stay. Comorbidities included: arterial hypertension, cardiovascular diseases, diabetes, obesity (defined as BMI > 30 kg/m^2^), chronic renal disease (stage 3–5 of CKD), hemodialysis, chronic pulmonary diseases (which included chronic obstructive pulmonary disease (COPD), bronchial asthma, emphysema or other pulmonary diseases), neoplasm during the last 5 years (which includes solid neoplasia or hematological malignancy), chronic neurological disorders, autoimmune diseases, obesity (defined as BMI > 30 kg/m^2^), and other diseases. Data acquisition and analysis were performed in compliance with protocols approved by the Ethical Committee of the National Institute for Infectious Diseases IRCCS Lazzaro Spallanzani, Rome, Italy (ethical approval number 164, 26 June 2020).

### 2.3. Statistical Analysis

Quantitative variables are expressed as medians (interquartile range, IQR), means (±standard deviation, SD), and 95% confidence intervals (95% CI). Nominal data are expressed as number (percentages, %). The statistical comparison was performed on mean values through the Mann–Whitney test for continuous variables and the Chi-Square test (or by the Fisher or Chi-Square test for trends where necessary) for categorical variables. Univariate and then multiple logistic regression analyses were completed. Statistical confounding factor selection was performed through backward elimination, abolishing from the model all nonsignificant confounders (*p*-value > 0.10). The adjusted Odds Ratio (aOR) values and 95% CI were reported. The statistical analysis was performed at the IRCCS L. Spallanzani Institute of Epidemiology, Rome, using the statistical software SPSS version 27 (IBM Corp. IBM SPSS, Armonk, NY, USA: IBM Corp).

## 3. Results

From 1 October 2020 to 31 December 2020, a total of 193 COVID-19 patients were admitted to our intensive care unit (ICU). Thirty-three patients with incomplete blood ammonia tests and four patients with liver disease were excluded. Therefore, a total of 156 ICU COVID-19 patients were included in the study. Of these, 19 patients had non-hepatic hyperammonemia (12.2%), of which 9 had low-NHH and 11 had high-NHH—Figure 1. The incidence of non-hepatic related hyperammonemia in our COVID-19 intensive care unit during 3 months of study was 12.2%. 

The median age of the cohort was 68.5 years (IQR, 58–76), 73.7% were male, and the median body mass index (BMI) was 27.7 (IQR, 25.4–31). The comorbidities with the highest prevalence were arterial hypertension (49.4%), obesity (36.5%), cardiovascular diseases (23.7%), diabetes (15.4%) and chronic lung diseases (14.7%)—Table 1.

From the 19 patients with NHH, 2 were treated with valproic acid, 1 with carbamazepine, 1 with positive rectal swab for Clostridium Difficile and 3 had bloodstream infection (2 patients with *Klebsiella* spp. and 1 with *Escherichia coli*). 

### 3.1. Risk Factors for NHH

The comparison in demographics, baseline characteristics and outcomes for the COVID-19 patients are summarized in Table 2.

In the univariate analysis, the statistically significant factor associated with the NHH patients was invasive mechanical ventilation, with an almost 6.6-fold higher risk (OR 6.66, 95% CI 0.86–51.6, *p* = 0.039). Age, sex, BMI, SOFA score and APACHE score at ICU admission; the presence of arterial hypertension, cardiovascular diseases, diabetes, obesity, chronic renal disease, hemodialysis or chronic pulmonary diseases; history of neoplasm, chronic neurological disorders, autoimmune diseases and other diseases were not significantly associated with NHH.

The multiple regression analysis showed that invasive mechanical ventilation had an independent 7-fold higher risk for NHH, but it not statistically significant (OR 7.1, 95% CI 0.90–56.4, *p* = 0.062).

Forty-nine patients (31.4%) in the cohort and five patients (26.3%) in the NHH group died within 28 days following ICU admission. No significant differences were observed in mortality at 28 days from ICU admission, in overall ICU mortality and also in ICU lengths of stay of more than 28 days.

### 3.2. Low-NHH vs. High-NHH

Comparing demographics, eight patients had blood ammonia levels classed as low NHH = 55–100 µmol/L and 11 patients had >100 µmol/L (high-NHH). No significant differences between the two groups were observed in age, sex, BMI, SOFA score and APACHE II score, Table 3. In addition, there was no significant differences in comorbidities between the two groups.

Furthermore, no statistically significant difference was found in the mortality between low-NHH (3 patients, 37.5%) and high-NHH (2 patients, 18.2%; *p* = 0.373) within 28 days from ICU admission or in ICU lengths of stay of more than 28 days.

## 4. Discussion

Testing for blood levels of ammonia in intensive care is usually performed in patients with prolonged or failed weaning, in those who have new onset seizures or with worsening neurological conditions [7]. To assess the incidence of hyperammonemia in our population, all patients were tested twice a week during the first 2 weeks from ICU admission, and in cases of hyperammonemia, they were monitored with tests every day until their values normalized.

The incidence of non-hepatic related hyperammonemia in our COVID-19 intensive care unit during the 3 months of study was 12.2%.

Invasive mechanical ventilation resulted in a significantly higher risk for NHH in the univariate regression, but it was not significant in the multivariate analysis. Age, sex, BMI, severity scores at ICU admission (SOFA score and APACHE score) and other comorbidities were not significantly associated with NHH. Furthermore, there were no significant differences in mortality between patients with and without hyperammonemia.

The question that could spontaneously arise is: Is there a correlation between COVID-19 and non-hepatic hyperammonemia, and should we be worried about NHH in these patients?

The exact mechanism of elevated ammonia levels in COVID-19 patients is not yet understood. In our study, we did not investigate all causes of non-hepatic-related hyperammonemia, such as any enzymatic dysfunctions of the urea cycle.

However, knowing some of the common predisposing conditions accepted by the scientific community for hyperammonemia [15,16], we can only hypothesize the probable causes or predispositions of hyperammonemia in some of our NHH patients. Medications such as valproic acid and carbamazepine can disrupt the urea cycle and can cause elevated ammonia levels; indeed, in the NHH group, two patients were treated with valproic acid and one patient with carbamazepine. In these cases, we can only hypothesize a presumed correlation; also, because not all of the patients treated with these medications presented with hyperammonemia, it was not clear if they were the real cause of hyperammonemia. Despite all of this, our first therapeutic step was to suspend or replace these medications. However, we know that infections caused by urease-producing bacteria can cause hyperammonemia, and in our NHH group, three patients had blood stream infections—of which two had *Klebsiella* spp. and one had *Escherichia coli*. Furthermore, in the ICU, increased muscle catabolism is frequent due to both persistent fevers and the immobility of patients—in which, as we all know, the striated muscular apparatus metabolizes excess ammonium into glutamine [4].

In the simultaneous presence of several of the aforementioned contributing causes, the amount of ammonium can exceed the ability of the liver to metabolize it, and this causes hyperammonemia that is not related to liver diseases.

Among the possible causes of hyperammonemia in COVID-19 patients, such as the persistence of a hyperinflammatory state, a predisposition to bacterial infections may be responsible, or a hypercoagulative state could be responsible for systemic micro-thromboembolisms—including intestinal ones [17]—that can cause the alteration of the intestinal mucosa and bacterial flora and a predisposition to intestinal infections caused by urease-producing bacteria [6].

A therapy aimed at reducing ammonium in the blood should be started as soon as possible. In cases with a known or suspected cause of hyperammonemia—such as drug therapy (valproic acid in two patients or carbamazepine in one patient), bacterial gastrointestinal infections (one patient with positive rectal swab for Clostridium Difficile) or sepsis (three patients)—the first approach, respectively, was to discontinue drug therapy, replace antiepileptic therapy with levetiracetam and to resolve the gastrointestinal infection or sepsis with targeted antibiotic therapy.

On the other hand, non-absorbable disaccharides are considered a first-line therapy—such as lactulose by nasogastric tube and/or by enemas—that is completely metabolized in the colon by bacterial flora, causing acidification in the colon that inhibits the growth of urease-producing bacteria, and also causing an increase in the osmotic pressure that expels ammonium out of the gut [18]. In addition, rifamixin is used as a gastro-intestinal (GI) site-specific antibiotic that is very effective in the treatment of hyperammonemia. Furthermore, sodium benzoate decreases blood ammonia levels by reducing glycine metabolism in the liver, kidney and brain; sodium phenylacetate/phenylbutyrate conjugates with glutamine in the liver and is then excreted by the kidneys; l-Arginine and l-Citrulline are products in the urea cycle and increase the elimination of ammonia [6].

We treated all patients with lactulose by nasogastric tube for at least 1 week. In addition, all patients with high NHH were treated with lactulose by enemas and rifamixin by nasogastric tube, until ammonia blood levels were normal. All patients with hyperammonemia were successfully treated without further complications.

During the weaning process, all 19 NHH COVID-19 patients performed a detailed daily neurological physical examination, and in cases of suspected encephalopathy, brain magnetic resonance imagery was performed to detect the characteristic lesions in the basal nucleus region (hyper-intensity in T1w and T2w). Brain magnetic resonance imagery was performed on three patients, but none of them showed hyperammonemia brain lesions. The hyperammonemia in all 19 COVID-19 patients resolved before ICU discharge, and none of them had neurological changes due to NHH during the weaning process.

This study has several limitations: First, this is a 3-month observational study designed in a single third-level COVID-19 center and is meant to be hypothesis-generating. Second, controlling for confounders, as in any observational study, may be incomplete despite all efforts, and the small sample size cannot provide precise estimates for predicted factors—which is one reason why the results of this study need to be confirmed in further research with larger studies. However, this study also has some strengths: observational studies are the first step in formulating a hypothesis that could incentivize the design of larger studies. This study observed COVID-19 patients over 3 months in a single third-level COVID center and shows a full representation of the real-life management of ICU COVID-19 patients with severe respiratory failure.

## 5. Conclusions

In conclusion, the incidence of NHH in ICU COVID-19 patients was not low, and in some patients at risk it is necessary to evaluate their levels of ammonia in order to treat them appropriately and in time. With all the limitations of the study, there was no correlation between NHH and demographics or clinical characteristics in COVID-19 patients in the ICU. The detailed pathogenesis of NHH in these patients may be a topic for further research.

## Figures and Tables

**Figure 1 jcm-11-02592-f001:**
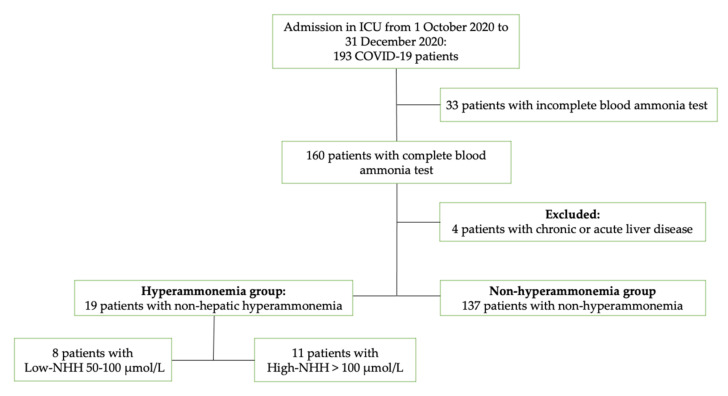
Flowchart of study selection; Abbreviations: COVID-19, Coronavirus disease 2019; ICU, intensive care unit; NHH, non-hepatic hyperammonemia.

**Table 1 jcm-11-02592-t001:** The baseline characteristics of the ICU COVID-19 patients.

	ICU COVID-19 Patients	NHH Group	Non-NHH Group
Characteristics	156	19	137
Age, median (IQR)	68 (58–76)	68 (60–76.5)	68 (58–76)
Male, *n* (%)	115 (73.7)	15 (79)	100 (73)
Female, *n* (%)	41 (26.3)	4 (21)	37 (27)
BMI, median (IQR)	27.7 (25.4–31)	27.3 (24.3–29)	27.8 (25.7–31)
SOFA score, median (IQR)	4 (3–7)	5 (3–6.5)	4 (3–7)
APACHE II score, median (IQR)	12 (8–18)	11 (8.5–15)	12 (8–18.2)

Abbreviations: COVID-19, Coronavirus disease 2019; ICU, intensive care unit; NHH, non-hepatic hyperammonemia; IQR, inter-quartile range; BMI, body mass index; SOFA score, sequential organ failure assessment and APACHE II score, acute physiologic and chronic health evaluation at ICU admission.

**Table 2 jcm-11-02592-t002:** Unadjusted and adjusted risk factors for NHH in ICU COVID-19 patients.

	ICU Patients	NHH Group	Univariate		Multivariate	
			OR (95% CI)	*p*	aOR (95% CI)	*p*
	156	19 (12.2)				
Age > 70 years old	67	8 (11.9)	0.96 (0.36–2.54)	0.937		
Male (vs. female), *n* (%)	115	15 (13)	1.39 (0.43–4.45)	0.581		
SOFA score ≥ 4	86	13 (19)	1.9 (0.68–5.29)	0.214		
APACHE II score ≥ 10	97	12 (12.4)	1.05 (0.39–2.83)	0.925		
Comorbidities, *n* (%)						
Arterial hypertension	77	12 (15.6)	1.89 (0.7–5.1)	0.199		
Cardiovascular diseases	37	7 (18.9)	2.08 (0.75–5.75)	0.151		
Diabetes	24	4 (16.7)	1.56 (0.47–5.18)	0.465		
Obesity ^a^	57	5 (8.8)	0.58 (0.19–1.71)	0.323		
Chronic renal disease ^b^	6	1 (16.7)	1.47 (0.16–13.3)	0.732		
Hemodialysis	5	1 (20.0)	1.85 (0.19–17.4)	0.587		
Chronic lung disease	23	6 (26.1)	2.51 (1.09–9.71)	0.103		
Previous neoplasm ^c^	13	2 (15.4)	1.35 (0.27–6.60)	0.712		
Chronic neurological disorders	11	2 (18.2)	1.67 (0.33–8.4)	0.528		
Autoimmune diseases	13	2 (15.4)	1.35 (0.27–6.6)	0.712		
Other chronical diseases	22	3 (13.6)	1.16 (0.31–4.38)	0.822		
Clinical characteristics						
Pre-ICU hospitalization ≥ 7 days	44	4 (9.1)	0.65 (0.20–2.07)	0.460		
PaO_2_/FiO_2_ ≤ 100 mmHg	16	4 (25)	2.78 (0.79–9.71)	0.098		
Invasive mechanical ventilation	118	18 (15.3)	6.66 (0.86–51.6)	0.039	7.1 (0.90–56.4)	0.062
ICU length of stay > 28 days	55	8 (14.5)	1.39 (0.52–3.69)	0.505		
28-day ICU mortality	49	5 (10.2)	0.75 (0.25–2.23)	0.610		
Overall ICU mortality	57	6 (10.5)	0.778 (0.28–2.2)	0.632		

Abbreviations: IQR, interquartile range; ICU, intensive care unit; BMI, body mass index; SOFA score, sequential organ failure assessment and APACHE II score, acute physiologic and chronic health evaluation at ICU admission; ^a^ obesity is defined as BMI > 30 kg/m^2^; ^b^ stage 3–5 of CKD, chronic kidney disease; ^c^ includes solid neoplasia or hematological malignancy in the last 5 years.

**Table 3 jcm-11-02592-t003:** Comparison between low-NHH and high-NHH in ICU COVID-19 patients.

	NHH Group (19)	L-NHH	H-NHH	*p*
No.	19	8	11	
Age, median (IQR)	68 (60–76.5)	71 (58.5–76.5)	66 (61.5–73.5)	0.868
Male, *n* (%)	15 (79%)	6 (75)	9 (81.8)	0.737
Female, *n* (%)	4 (21%)	2 (25)	2 (18.2)	0.737
BMI, median (IQR)	27.3 (24.3–29)	27.7 (26.8–27.7)	26.8 (23.1–28.2)	0.078
SOFA score, median (IQR)	5 (3–6.5)	5 (3–6)	5 (3.5–6.5)	0.883
APACHE II score, median (IQR)	11 (8.5–15)	12 (8.7–17.2)	11 (8.5–14)	0.579
Comorbidities, *n* (%)				
Arterial hypertension	12 (63.1%)	6 (75)	6 (45.5)	0.390
Cardiovascular diseases	7 (36.8%)	4 (50)	3 (27.7)	0.338
Diabetes	4 (21.0%)	2 (25)	2 (18.2)	0.737
Obesity ^a^	5 (26.3%)	3 (37.5)	2 (18.2)	0.373
Kidney disease ^b^	1 (5.3%)	1 (12.5)	0 (0)	N.A.
Hemodialysis	1 (5.3)	1 (12.5)	0 (0)	N.A.
Chronic lung disease	6 (31.6%)	2 (25)	4 (36.4)	0.623
Previous neoplasm ^c^	3 (15.7%)	1 (12.5)	2 (18.2)	0.824
Chronic neurological disorders	2 (10.5%)	0 (0)	2 (18.2)	N.A.
Autoimmune diseases	2 (10.5%)	1 (12.5)	1 (9.1)	0.824
Other chronical diseases	3 (29.0%)	2 (25)	1 (9.1)	0.376
Clinical characteristics, no. (%)				
Pre-ICU hospitalization ≥ 7 days	4 (21)	0 (0)	4 (36.4)	N.A.
PaO_2_/FiO_2_ ≤ 100 mmHg	4 (21)	3 (37.5)	1 (9.1)	0.149
Invasive mechanical ventilation	18 (94.7)	7 (87.5)	11 (100)	N.A.
ICU length of stay > 28 days	8 (42)	3 (37.5)	5 (45.5)	0.746
ICU mortality, no (%)				
28-day ICU mortality	5 (26.3)	3 (37.5)	2 (18.2)	0.373
Overall ICU mortality	6 (31.6)	3 (37.5)	3 (27.3)	0.658

Abbreviations: COVID-19, Coronavirus disease 2019; ICU, intensive care unit; NHH, non-hepatic hyperammonemia; IQR, inter-quartile range; BMI, body mass index; SOFA score, sequential organ failure assessment and APACHE II score, acute physiologic and chronic health evaluation at ICU admission; ^a^ obesity is defined as BMI > 30 kg/m^2^; ^b^ stage 3–5 of CKD, chronic kidney disease; ^c^ includes solid neoplasia or hematological malignancy in the last 5 years; N.A., not applicable.

## Data Availability

The data presented in this study are available on request from the corresponding author. The data are not publicly available be-cause of patient privacy and data protection regulations.

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
