# Peer review of "Epidemiology, Clinical Presentation and Treatment of Non-Hepatic Hyperammonemia in ICU COVID-19 Patients"

_jcm, 2022, doi:10.3390/jcm11092592_

Round 1

Reviewer 1 Report

Nice work ! I congratulate !

I think it would be useful for the readers to have 1-2 sentences about NHH in non-COVID-19 ICU patients in general and in those that are ventilated due non-COVID reasons - there is date - just that the reader can fell the problem.

NHH is well known in patients with septic; you state that you had (only) three patients with infections out of 19 that developed NHH; here it would be nice to have the information on how many of pts that did not develop NHH had documented bacterial infections? Were infections included in the analysis? 

COVID-19 ICU patients can rather often bleed from upper GI tract but also from intestines, and clinically rather silent colitis could be found more in the search for bleeding source. Considering the pathophisiology - connection of GI bleeding and NHH, I suggest to give the data on GI bleeding - manifest or if erythrocytes needed to be supplemented

Some sentences in the abstract and main text are difficult to understand and the English language flow is not good - could be rewritten to be more clear.

Author Response

Dear Reviewer,

we thank you for your valuable suggestions. We took your advice and made changes on the manuscript.

Below we will try to answer your questions as best we can:

“I think it would be useful for the readers to have 1-2 sentences about NHH in non-COVID-19 ICU patients in general and in those that are ventilated due non-COVID reasons - there is date - just that the reader can fell the problem.”

  • Done, check please lines 67-68, the most common causes of hyperammonemia reported in the medical literature have been described on lines 221-237.

“NHH is well known in patients with septic; you state that you had (only) three patients with infections out of 19 that developed NHH; here it would be nice to have the information on how many of pts that did not develop NHH had documented bacterial infections? Were infections included in the analysis?”

  • We do not have that analysis because the authors considered it irrelevant in the comparison between NHH and non-NHH, because there were few cases of sepsis.

“COVID-19 ICU patients can rather often bleed from upper GI tract but also from intestines, and clinically rather silent colitis could be found more in the search for bleeding source. Considering the pathophisiology - connection of GI bleeding and NHH, I suggest to give the data on GI bleeding - manifest or if erythrocytes needed to be supplemented”

  • We have had no known cases of gastrointestinal bleeding in this case-series.

“Some sentences in the abstract and main text are difficult to understand and the English language flow is not good - could be rewritten to be more clear”

  • We made some changes in the abstract and in main text. All revisions to the manuscript are flagged using the "Track Changes" function of MS Word, such that any changes can be easily viewed.

Please feel free to contact me if you have any question.

Kind regards,

Dr. Nardi Tetaj

Reviewer 2 Report

First of all, this study is novel. I have not seen this kind of study in COVID-19. Although further studies are needed to find the cause, this study has some clinical meanings. I have some minor comments.

How did you dcide the cut-off point of hyperammonemia level. Please state the reason why you decided it is 50.  For example, previous lieteratures or ROC curve.

What is the average level of ammonemia in critically ill patients in general? Please compare it with your data in COVID-19.

You mentioned some medications may have affected the hyperammonemia in COVID 19. How many people in your patients used such kind of drugs?

Please indicate when I need to check ammonia in COVID-19 patients? Only in mechanically ventilated patients? When? daily? 

Did you treat the hyperammonemia in COVID-19? How was the response?

How was the trend of hmmonia in critically ill COVID-19 patients?

Author Response

Dear Reviewer, we thank you for your valuable suggestions. We took your advice and made changes on the manuscript.

Below we will try to answer your questions as best we can:

“How did you decide the cut-off point of hyperammonemia level. Please state the reason why you decided it is 50.  For example, previous lieteratures or ROC curve.”

  • The references of the laboratory values were based on the literature you find in the book: “Ammonia. Mosby's Diagnostic and Laboratory Test Reference. 13th ed. St. Louis, Mo: Elsevier; 2017” and on https://emedicine.medscape.com/article/2054408-overview.

“What is the average level of ammonemia in critically ill patients in general? Please compare it with your data in COVID-19”

  • The incidence of ammonemia in critically ill patients reported in the medical literature is still controversial.

“You mentioned some medications may have affected the hyperammonemia in COVID 19. How many people in your patients used such kind of drugs?”

  • Check the lines 221-228. In our NHH group, 2 patients were treated with valproic acid and 1 patient with carbamazepine. Our first therapeutic step was to stop or replace these drugs, although it was unclear whether they were the real cause of the hyperammonemia. However, we also had many other patients treated with these drugs but who did not have hyperammonemia, but who were not included in the statistical analysis because the authors considered it irrelevant for the aforementioned reason.

“Please indicate when I need to check ammonia in COVID-19 patients? Only in mechanically ventilated patients? When? daily?“

  • Check the lines 202-207. Testing for blood levels of ammonia in intensive care is usually performed in patients with prolonged or failed weaning, in those who have new onset seizures or with worsening neurological conditions. All patients in ICU were tested twice a week during the first 2 weeks from ICU admission, and in case of hyperammonemia they were monitored by testing every day until their values normalized.

“Did you treat the hyperammonemia in COVID-19? How was the response?”

  • Please check the lines 261-264. We treated all patients with lactulose by nasogastric tube for at least 1 week. In addition, all patients with high NHH were treated with lactulose by enemas and rifamixin by nasogastric tube, until ammonia blood levels were normal. All patients with hyperammonemia were successfully treated without further complications.

“How was the trend of ammonia in critically ill COVID-19 patients?”

  • After the start of treatment, the blood levels of ammonia of all patients was improving

All revisions to the manuscript are flagged using the "Track Changes" function of MS Word, such that any changes can be easily viewed.

Please feel free to contact me if you have any question.

Kind regards,

Dr. Nardi Tetaj
